# The Effect of Delivery Matrix on *Bifidobacterium animalis* subsp. *lactis* HN019 Survival through In Vitro Human Digestion

**DOI:** 10.3390/nu15163541

**Published:** 2023-08-11

**Authors:** Nicolas Yeung, Sofia D. Forssten, Markku T. Saarinen, Mehreen Anjum, Arthur C. Ouwehand

**Affiliations:** IFF Health & Nutrition, Sokeritehtaantie 20, 02460 Kantvik, Finland; sofia.forssten@iff.com (S.D.F.); markku.saarinen@iff.com (M.T.S.); mehreen.anjum@iff.com (M.A.); arthur.ouwehand@iff.com (A.C.O.)

**Keywords:** probiotic, microbiota, digestion simulation, qPCR, short-chain fatty acid, branched-chain fatty acid, in vitro

## Abstract

*Bifidobacterium animalis* subsp. *lactis* HN019 is a probiotic with several documented human health benefits. Interest in probiotics has led to the development of new formats that probiotics, including HN019, can be supplemented into. In this study, we looked at common HN019 formats such as frozen culture and freeze-dried powder as well as supplementing it into the following food matrices: yogurts (dairy, soy, and oat based), xanthan gum-based tablets, pulpless orange juice, whey sports drink, and dark chocolate (70% cocoa). In this work, our aim was to investigate whether the food matrix that carried HN019 via simulated human digestion (a dual model system mimicking both upper and lower gastrointestinal digestion) influenced probiotic delivery. To that end, we validated and used a real-time qPCR assay to detect HN019 after simulated digestion. In addition, we also measured the effect on a panel of metabolites. After simulated digestion, we were able to detect HN019 from all the matrices tested, and the observed changes to the metabolite profile were consistent with those expected from the food matrix used. In conclusion, this work suggests that the food matrix supplemented with HN019 did not interfere with delivery to the colon via simulated human digestion.

## 1. Introduction

*Bifidobacterium animalis* subsp. *lactis* HN019 (HN019) has a long history of use as a probiotic, and its health benefits have been demonstrated in numerous human clinical trials. These trials have investigated HN019′s ability to modulate the immune system [1], improve colonic transit time and constipation [2] and reduce childhood morbidity in the developing world [3] when given with a prebiotic in fortified milk [4]. Probiotics, by definition, are ‘live microorganisms that, when administered in adequate amounts, confer a health benefit on the host’, with the health benefits being strain specific to the microorganism [5]. Given the rise of public interest in the link between the human gut microbiota and health, as well as the increase in finished food formats that can be supplemented with probiotics, it is essential not only to consider the strain of probiotic but also the vehicle of delivery to the host, which this study aims to investigate [6,7].

In vitro models that can simulate human gastrointestinal (GI) digestion are useful investigative tools for screening the potential effects of a food matrix on probiotic delivery to the gut, providing additional information prior to larger investments in human clinical trials. In this study, we combined two models that simulate human GI digestion in order to examine common finished formats that probiotics are commercially produced in (post-fermentation frozen culture pellets and freeze-dried culture powder) as well as some common finished formats that probiotics are supplemented into the following: dairy yogurt as well as soy- and oat-based ‘yoghurts’, 70% dark chocolate (chocolate), xanthan gum-based tablets (tablets), and beverages such as fruit juice and whey sports drink. With the aim to measure the effect of the food matrix on the ability to recover genomes of the studied probiotic organism, HN019, after GI simulation, which we are using as a proxy for delivery to the human gut and metabolic effects.

The four-stage colon simulator used in this study has also been previously used to investigate similar food matrices, such as prebiotics containing cocoa mass [8] and probiotic cheese [9]. Given the colon simulator’s ability to screen complex food matrices and coupling it with a pre-digestion model designed to mimic upper GI digestion, this creates a unique platform to investigate the effect of food matrix on HN019 delivery to and survival in the human gut.

HN019 has been administered, historically during (pre)clinical trials, in various formats; freeze-dried powder [3], capsules [10], lozenges [11], low-fat or fat-free milk [12,13,14], and yogurt [15,16,17]. Capsules with freeze-dried microbes are the most common delivery matrix for probiotics in the current literature.

Various molecular techniques are available to analyze the microbiota from fecal material collected during clinical trials in which probiotics have been consumed. Sequencing-based methods can be used to observe changes across the microbiome. However, this lacks the resolution to detect the presence or absence of probiotics at the species or strain level. Strain-specific real-time qPCR assays are the standard tool for detecting the delivery of probiotics to the human gut by recovering probiotic DNA from fecal samples. With the abundance of bacterial genetic information being produced worldwide, it is also important to update and validate strain-specific PCR primers and probes. Here, a previously reported set of HN019-specific primers and probes were utilized to quantify HN019 after administration in various delivery formats from simulated human GI transit.

Taken together, the aim of this study was to investigate the effect of the delivery matrix on HN019 via simulated human digestion using modern molecular approaches, comparing common and novel matrices. 

## 2. Materials and Methods

### 2.1. Ethics

This study and all methods used in it were carried out in accordance with relevant guidelines and regulations. The study was not considered medical research as defined in the Finnish Act on Medical Research (488/1999, as amended). Hence it did not require approval from the ethical committee and such approval was not obtained. In addition, as the study was not considered as medical research, the informed consent has not been obtained in writing from all research subjects, as required by the Act on Medical research, but orally. 

### 2.2. Food Matrices

The matrices selected for these simulations were as follows: dairy yogurt, non-dairy soy ‘yoghurt’, non-dairy oat ‘yoghurt’, pulpless orange juice, whey protein sports drink, xanthan gum-based tablet, dark chocolate (70% cocoa), and common formats of freeze-dried and frozen culture of HN019 (Table 1). The freeze-dried, frozen culture and dark chocolate were products from Danisco USA (Madison, WI, USA). The xanthan gum-based tablets (ProBion^®^) were provided by Wasa Medicals (Halmstad, Sweden). The dairy yogurt, not containing probiotics, was an unflavored product (Bulgarian Yoghurt, Valio, Helsinki, Finland), inoculated using frozen HN019 culture. The same approach was used for the following: pulpless orange juice (Tropic Pulpless, Eckes-Granini, Turku, Finland), non-dairy natural flavored soy ‘yoghurt’ (Alpro, Wevelgem, Belgium), non-dairy vanilla flavored oat ‘yoghurt’ (Yosa, Fazer, Helsinki, Finland), and the chocolate flavored whey protein sports drink (Gainomax, Midsona, Malmö, Sweden).

### 2.3. Upper Gastro-Intestinal (GI) Digestion and Colon Simulation

The upper GI and colon simulations were conducted as described in [9]. In short, 15 mL (or grams in the case of a solid matrix) of matrix, inoculated with probiotic to total amount shown in Table 1, was kept at 37 °C (water bath) with constant magnetic stirring. Buffers were added in succession as follows: saliva (pH 6.5, with α-amylase, uric acid, and mucin) for 5 min; gastric (pH 2.5, with BSA, pepsin, and mucin) for 75 min; duodenal (pH 6.5, with BSA, pancreatin and lipase) and bile (pH 6.5, with BSA and bile) for 90 min each. To mimic absorption and remove soluble digested material, the volume was then centrifuged at 10,000× *g* for 20 min at room temperature, and the resulting pellet was further processed in an anaerobic chamber where it was mixed with 50 mL of synthetic ileal medium [18] and carried forward into the 4-stage colon simulation. Colonic simulation was performed as described in [19]. Fecal inoculates for the colon simulations were donated by healthy subjects, not consuming probiotics on regular basis for the last 3 months nor taking antibiotics for the last 3 months. The fresh fecal samples were processed directly for the colon simulations [19].

In addition to the matrices, a control not containing probiotics was digested in the same manner. All downstream analyses were performed on all test conditions and the control. This was to normalize for donor microbiota variations and any possible background, particularly for any other HN019-like organisms. At the completion of each simulation, the vessels were individually sampled. The samples were stored at −80 °C until analyzed.

### 2.4. DNA Extraction

The bacterial DNA was extracted from the material collected from the simulation vessels using an initial bead beating step of two 3 × 30 s cycles at 6800 rpm with a Precellys bead beater (Bertin Instruments, Montigny-le-Bretonneux, France), where, after, the DNA was purified and extracted with an automated MagMAX™ Sample Preparation System (Life Technologies, Halle, Belgium), using the MagMAX™ Nucleic Acid Isolation Kit. The quantity of extracted DNA was determined using a Qubit^®^ dsDNA HS Assay Kit (Thermo Fisher Scientific, Vantaa, Finland), as described previously [20].

### 2.5. qPCR: Bifidobacterium animalis subsp. lactis HN019 Primer Design

There are several strains of *Bifidobacterium animalis* subsp. *lactis* (e.g., HN019, B420, Bl-04, and Bi-07) that share near identical genetic identity [21]. This makes strain-specific primer design challenging, but a set of primers and a probe were designed [22], targeting a region of CRISPR-cas repeats [23]. These inserted repeats create one amplicon of 158 base pairs (bp), which corresponds to strains HN019/B420, and 372 base pairs (bp), which corresponds to strains HN019/B420 and Bl-04/Bi-07, respectively (Appendix A).

### 2.6. qPCR: Assay Conditions

The B. lactis qPCR assay was performed using a 7500FAST Real-Time PCR instrument (Applied Biosystems) and TaqMan FAST advanced mastermix (ThermoFisher Scientific, Waltham, MA, USA) with 600 nM HN019_F (5′-TTCGATGGTTCGCACAGTGA-3′), 900 nM HN019_R (5′-GGTCTGATGCCGCCTGAAAT-3′) and 200 nM HN019_P (5′-FAM/AAACAGGTC/ZEN/AATCAGCGGCGCAGGGAG-3′) with an annealing temperature of 60 °C [22]. Triplicate reactions were carried out in a volume of 15 µL, of which 5 µL was sample DNA (0.2 ng/µL, giving 1 ng of total DNA). A standard curve was used for absolute quantification, which consisted of DNA extracted from pure cultured HN019. The DNA from the pure HN019 culture was extracted as described above. Similarly, as negative controls, other closely related strains (Appendix A) and NTC were used. The DNA from these was extracted from pure cultures. The standard curve started at 10 ng of total DNA and was serially diluted ten-fold down to 10 fg, while all negative controls were at the same input (1 ng) as the simulation samples. For samples that showed non-quantifiable values, a theoretical limit of detection (5.37 Log^genomes/mL^) was set at half the limit of quantification, which was the last standard of 10 fg. This limit of detection was used instead of zero. The unit of Log^genomes/mL^ was determined from the original qPCR value in genomes/ng of DNA, correlating it with the DNA concentration in ng of DNA/µL and then finally taking into account the dilution factors associated with sample handling post-simulation (i.e., bead beating and DNA extraction).

The simulations were conducted in two separate rounds that were separated for more than 18 months. The first round included the following matrices: freeze-dried and frozen culture HN019, dairy yoghurt, chocolate, and tablet. The second round comprised the following: soy and oat yogurts, whey sports drinks, and pulpless orange juice. Each round of simulation’s controls was compared to each other using a Mann–Whitney test and found to be significantly different. The first round had a control of 5.76 ± 0.60 (mean ± standard deviation) Log^genomes/mL^ versus the second round of 6.30 ± 0.59 Log^genomes/mL^, with a *p* value of 0.0024. The data sets were normalized for combined analysis.

### 2.7. Short-Chain, Branched-Chain Fatty Acid and Lactic Acid Quantification

Short-chain fatty acids (SCFA) [acetic acid, propionic acid, butyric acid, valeric acid], branched-chain fatty acids (BCFA) [isobutyric acid, 2-methylbutyric acid, isovaleric acid] and lactic acid were analyzed from the colon simulator samples with gas chromatography, as described previously [24] with modifications. An internal standard (0.1 mL 20 mM pivalic acid), 0.3 mL of water, and 0.25 mL of saturated oxalic acid solution were added to 0.1 mL of sample. After thorough mixing, the sample was incubated for 60 min at 4 °C and then centrifuged at 16,000× *g* for 5 min at room temperature. Supernatant (1 µL) was analyzed via gas chromatography using a glass column packed with 80/120 Carbopack BDA/4% Carbowax 20 M stationary phase (2 m × 2 mm, Supelco, Bellefonte, PA, USA) at 175 °C and using helium as the carrier gas at a flow rate of 24 mL/min. The temperatures of the injector and the flame ionization detector were 200 °C and 245 °C, respectively.

### 2.8. Statistical Analysis

Statistics were performed using Graphpad Prism software version 9.5 (GraphPad Software Inc., San Diego, CA, USA). A one-way ANOVA test was performed using Kruskal–Wallis’ multiple comparisons test for non-parametric data, comparing each matrix to its respective control. Each matrix was run in at least 3 separate simulations, except for freeze-dried HN019, which was run twice. The data are shown as a pool of all four vessels, except for one figure looking specifically at lactic acid and the chocolate matrix. Statistical significance in the figures was left default in Prism [(Ns = *p* > 0.05) (* = *p* ≤ 0.05) (** = *p* ≤ 0.01) (*** = *p* ≤ 0.001) (**** = *p* ≤ 0.0001)]. qPCR data were computed in their Log^genomes/g^ feces unit, while short-chain fatty acid, fatty acid, and lactic acid were analyzed in µmol/mL.

## 3. Results

### 3.1. HN019 DNA Recovery

All the matrices, except dairy yogurt, were statistically higher than the control (Figure 1). The matrices freeze-dried HN019, soy yoghurt, oat yoghurt, pulpless orange juice, and whey sports drink were able to increase from a control value of 5.81 ± 0.59 Log^genomes/mL^ with *p* values < 0.0001. The matrices chocolate, tablet, and frozen culture also showed an increase with *p* values of 0.006, 0.013, and 0.019, respectively. The dairy yogurt had a value of 6.01 ± 0.55 Log^genomes/mL^.

### 3.2. Short-Chain Fatty Acids

In Figure 2, the results of the SCFA analyses are represented. The oat yogurt had the largest overall change in total SCFA compared to the control, with a *p* value of 0.0003. Acetic acid was significantly increased in the oat yogurt (*p* value of 0.0058) and chocolate (*p* value < 0.0001) matrices. Propionic acid was significantly increased in both the soy and oat yogurts with *p* values < 0.0001, while both butyric and valeric acids were not significantly changed in any of the matrices compared to the control.

### 3.3. Branched-Chain Fatty Acids

The total amount of BCFAs was increased in the soy (*p* value of 0.038) and oat yogurt matrices (*p* value of 0.010) and decreased in the chocolate matrix (*p* value of 0.025, Figure 3). The oat yogurt increased the amounts of isobutyric (*p* value of 0.015), 2-methylbutyric (*p* value of 0.02), and isovaleric (*p* value of 0.01) acids, while the soy yogurt only increased the amount of isovaleric (*p* value of 0.04) acid. The chocolate matrix significantly lowered the amount of isobutyric acid (*p* value < 0.0001).

### 3.4. Lactic Acid

The chocolate matrix showed a significant increase (*p* value < 0.0001) of lactic acid from control (Figure 4a), which was not observed for the other tested matrices. A closer examination of the results from the chocolate matrix in Figure 4b shows a rise in lactic acid levels from the first to the second vessel, a stabilization of the lactate level in the third vessels, followed by a drop in the fourth and last vessel of the colon simulator.

## 4. Discussion

The interplay between a probiotic and its mode of delivery to and through a host’s digestive system is an important consideration when evaluating the potential health benefit provided by said probiotic. In this study, we aimed to observe the effect of various matrices, all carrying HN019, using a two-part simulated human digestion model. We combined a model for simulating upper GI digestion and fed that into a standalone simulator designed to model the colon in four separate stages, similarly to [9]. 

We extracted DNA after simulated digestion to monitor probiotic DNA recovery, which we have used as an approximation for transit through the human gastrointestinal tract. We acknowledge that, by definition, probiotics must be alive and that recovered DNA is not necessarily proof of viability. However, it is known that nucleic acids are metabolized throughout the human gastrointestinal tract. Therefore, any non-membrane-protected DNA would be readily degraded and, thus, not available for detection via PCR [25]. This degradation was shown to occur especially during gastric digestion via a mechanism that is thought to be driven by pepsin, which was a component of the upper GI digestion model used here. Strain-specific qPCR is the current standard in human clinical trials and has been successfully used [10,26]. To that end, we also validated a set of strain-specific primers and probes [22] that can be used in the detection and enumeration of strain HN019 from human feces. The assay detected both HN019 and B420 with greater than 90% PCR efficiency. It also had no amplification across a panel (Appendix A) of other commensal bacteria.

The results of probiotic DNA recovery across all tested matrices, except for the dairy yogurt, showed a significant increase from the control. The results suggest that the tested matrices, with the notable exception of yogurt, had no effect on the delivery of HN019 via simulated human digestion. This was unexpected as delivery of *B. lactis* in yogurt has been shown to result in fecal recovery [27]. Otherwise, the results agree with human studies as far as freeze-dried material (capsules) [10] and milk [4] are concerned. This suggests that intact bacteria were surviving both upper and lower GI digestion, which gives evidence that at least a portion of live bacteria were able to survive passage through the GI tract.

The most notable changes in SCFA were observed in simulations with the vegetable-based ‘yoghurts’, oat and soy, and chocolate. These matrices significantly increased the amounts of acetic and propionic acid compared to the control simulation. This is likely explained by the content of fiber [28,29,30] in these products, which escape digestion and are fermented in the colon by the resident microbiota. However, pulpless orange juice had no change from the control, which is in contrast to other studies that used whole orange juice and showed a significant increase in acetic, butyric, and propionic acids using a similar simulated digestion model [31]. This is likely due to the removal of soluble fiber in the pulpless orange juice used in this study.

In general, the abundance and, therefore, absolute measured amounts of BCFAs vs. SCFAs was lower. This finding is consistent with work conducted previously [8]. The increase in SCFA due to the increase in acetic acid is, at least for chocolate, correlating negatively with BCFA, mainly via a sharp reduction in isobutyric acid. This can be explained by the preferred use of fiber as an energy source by the intestinal microbiota leading to reduced fermentation of proteinaceous material and thereby generating less BCFA. The amount of BCFAs slightly increased in the vegetable-based oat and soy ‘yoghurts’, which might be explained by the relatively low amount of fiber and higher protein content of the plant-based milk used in their production [32]. Interestingly, dairy yogurt did not show this same mild increase in BCFAs despite having similar metabolomic characteristics [33].

The chocolate matrix had the largest impact on lactic acid levels compared to the other tested matrices, with a marked increase. Looking more closely, we can see that this increase was transient across the simulation vessels. Fermentable material that escaped the simulated digestion is likely to have been metabolized by the simulated colonic microbiota during the first two vessels in the colon model. Towards the end of the simulator, when carbon and energy sources are depleted, the lactate is further metabolized to SCFA, and the last vessel has little measurable lactate left. This stands in line with work looking into lactic acid fermentation in the human gut and its conversion to acetic acid [34].

Interestingly, even though the whey sports drink was chocolate flavored, the same increase was not seen. This could be due to the relatively low amount of cocoa present in the drink (1%) since it has been shown that a 3% amount of cocoa is needed to have a beneficial effect on host lactic-acid-producing bacteria [35].

## 5. Conclusions

Taken together, this work suggests that the effect of delivery vehicles on probiotic administration via simulated human digestion was a non-factor. Although, the metabolic effect of the vehicle itself should be considered, as it was shown to have an impact on the metabolite profile during simulated digestion. This is consistent with work previously carried out using this model. As the field of application science continues to broaden, the list of vehicle probiotics that can be administered, no doubt, will grow. This type of combined in vitro model can provide insights into how food matrices may influence probiotic survival in the GI tract. Furthermore, the matrices tested showed similar results in terms of probiotic passage through the simulated GI tract. Altogether, this work showed that the delivery matrix had no appreciable effect on probiotic passage through simulated human digestion. Though this work focused on probiotic DNA recovery as an approximation for intact bacteria, the field of molecular biology has expanded greatly in recent years. Future work could adopt methods involving digital PCR and a viability dye to further establish the link between PCR results and viable bacteria [36].

## Figures and Tables

**Figure 1 nutrients-15-03541-f001:**
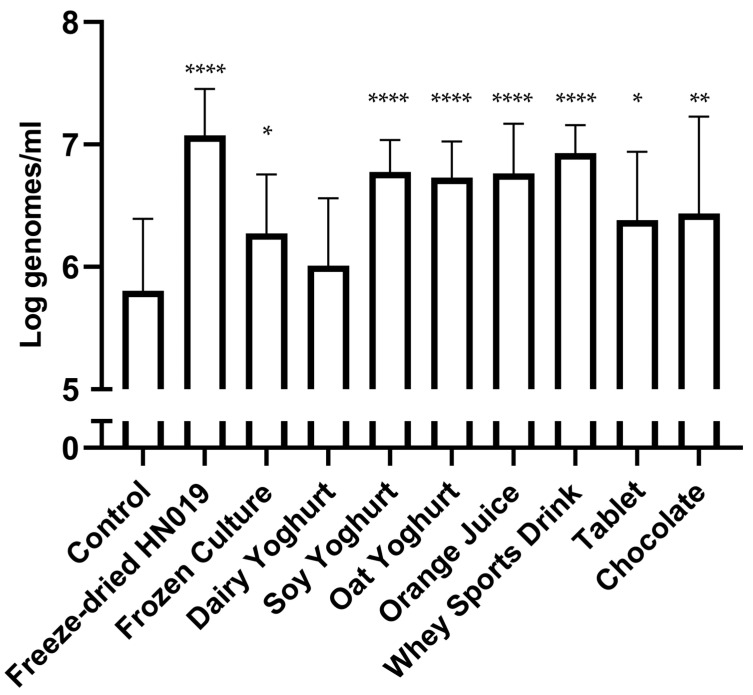
Probiotic recovery after simulated digestion through the upper GI tract and the colon. Control was a colon simulator subunit that had no test substance added to its feed vessel, controlling for donor microbiota variation per simulation. The *Bifidobacterium animalis* subsp. *lactis* HN019 qPCR results as log^genomes/mL^ + standard deviation represent the combined measurements across all four vessels of each simulator subunit. [(* = *p* ≤ 0.05) (** = *p* ≤ 0.01) (**** = *p* ≤ 0.0001)].

**Figure 2 nutrients-15-03541-f002:**
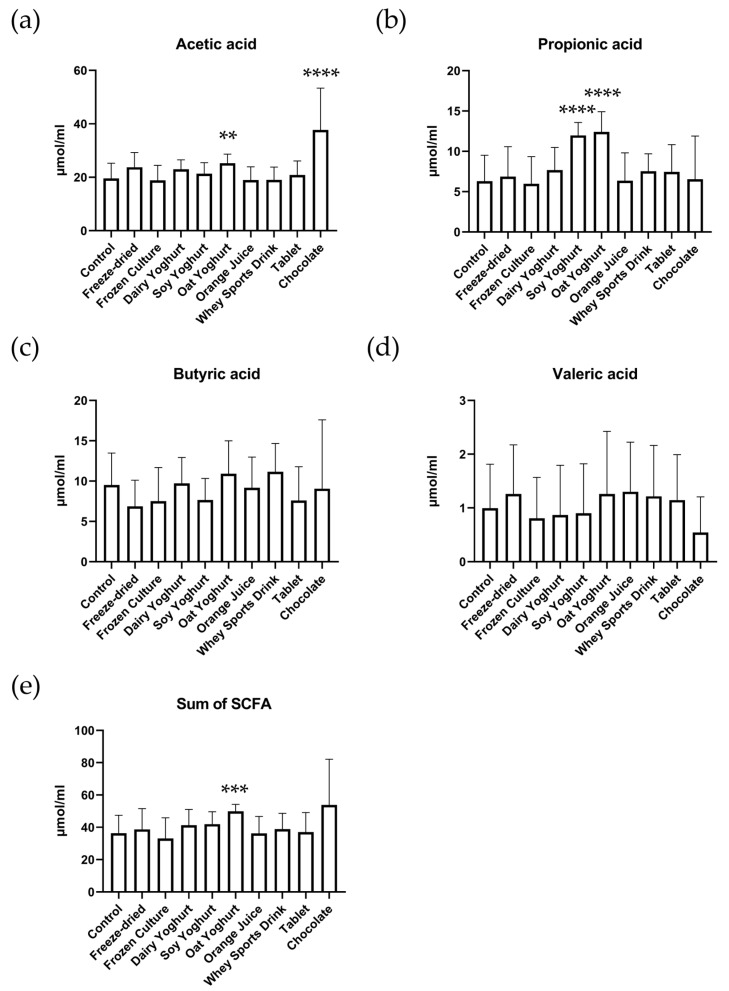
Short-chain fatty acids (SCFA) levels after simulated digestion through the upper GI tract and the colon. The levels of acetic (**a**), propionic (**b**), butyric (**c**), and valeric (**d**) acid and their sum (**e**). Data represent the average amount + standard deviation measured across all four vessels of each simulator subunit. [(** = *p* ≤ 0.01) (*** = *p* ≤ 0.001) (**** = *p* ≤ 0.0001)].

**Figure 3 nutrients-15-03541-f003:**
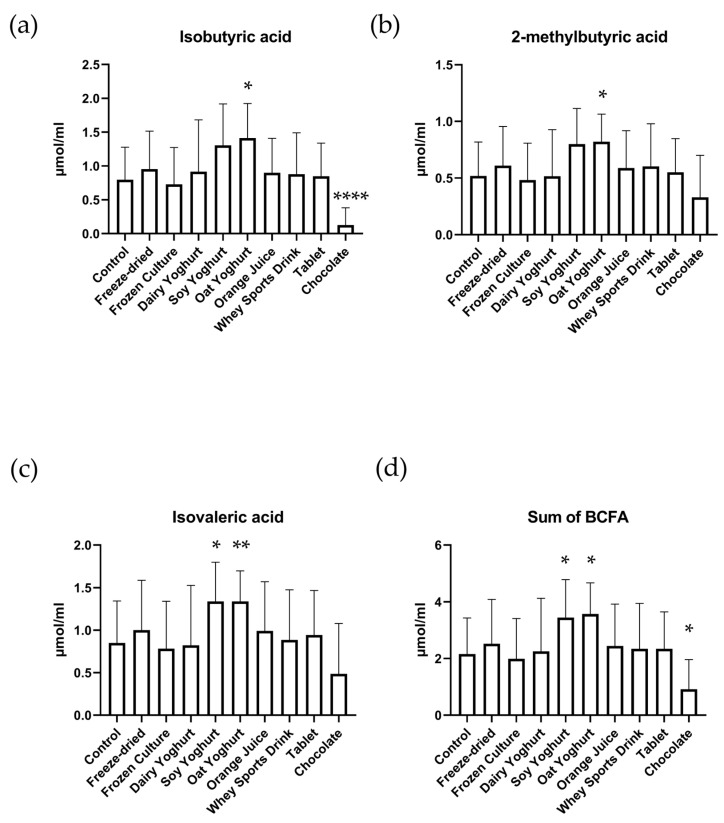
Branched-chain fatty acids (BCFA) levels after simulated digestion through the upper GI tract and the colon. The levels of isobutyric (**a**), 2-methylbutyric (**b**), and isovaleric (**c**) acid and their sum (**d**) represent the average amount + standard deviation measured across all four vessels of each simulator subunit. [(* = *p* ≤ 0.05) (** = *p* ≤ 0.01) (**** = *p* ≤ 0.0001)].

**Figure 4 nutrients-15-03541-f004:**
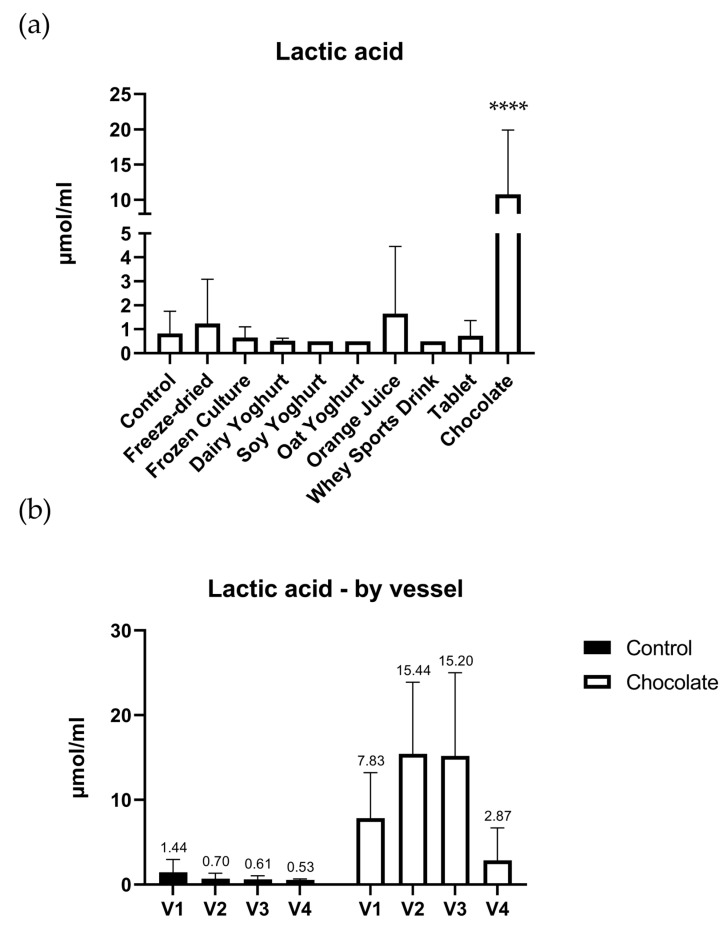
(**a**) Lactic acid levels after simulated digestion through the upper GI tract and the colon. The levels represent the average amount + standard deviation measured across all four vessels of each simulator subunit. (**b**) Lactic acid levels per vessel. [(**** = *p* ≤ 0.0001)].

**Table 1 nutrients-15-03541-t001:** Food matrix information with total amount of added *Bifidobacterium animalis* subsp. *lactis* HN019.

Food Matrix	Total CFU	Matrix Manufacturer	Other Major Ingredients
Frozen culture	2.025 × 10^9^	IFF (Madison, WI, USA)	none
Freeze-dried powder	2.025 × 10^9^	IFF (Madison, WI, USA)	none
Dairy yogurt	2.025 × 10^9^	Valio (Helsinki, Finland)	pasteurized milk, starter culture, vitamin D
Non-dairy soy yogurt	2.025 × 10^9^	Alpro (Wevelgem, Belgium)	soya base (water and hulled soya beans), sugar, tri-calcium citrate, pectins
Non-dairy oat yogurt	2.025 × 10^9^	Fazer (Helsinki, Finland)	water, oat, sugar, corn starch, locust bean gum
Chocolate	7.842 × 10^9^	IFF (Madison, WI, USA)	cocoa mass, sugar, cocoa butter, palm oil, vanilla (min. cocoa solids 70%)
Tablet	1.32 × 10^9^	Wasa Medicals (Halmstad, Sweden)	inulin, xanthan gum, magnesium stearate
Orange juice (pulpless)	2.025 × 10^9^	Eckes-Granini (Turku, Finland)	none
Whey sports drink	2.025 × 10^9^	Midsona (Malmö, Sweden)	UHT-treated milk, maltodextrin, milk protein, sugar, cocoa

## Data Availability

The data presented in this study are available upon request from the corresponding author. The data are not publicly available due to privacy.

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
