# Peer review of "The Effect of Delivery Matrix on *Bifidobacterium animalis* subsp. *lactis* HN019 Survival through In Vitro Human Digestion"

_nutrients, 2023, doi:10.3390/nu15163541_

Round 1
Reviewer 1 Report
The manuscript of Yeung and colleagues touches an important topic in the probiotic field, the matrix effect on probiotic delivery and GI transit survival. The manuscript is well written but it needs to be improved in some sections to allow the reader to clearly understand all technical issues and the results presented.
-Table 1 - The authors should specify to what amount of matrix the CFU refers. Are they CFU/g or total CFU in a defined amount of matrix.
- lines 107 -110 - Here a definition of "control" is reported. However it not clear which data are obtained from the "control". Is this "control" the same used in Figure 1? Please specify.
_lines 139-141 - The authors describe how they obtained the limit of quantification of the qPCR assay developed. Hovewer, the value of this limit is not reported. In addition the authors arbitrary fixed the limit of detection as half of the value of the limit of quantification, whereas it should be calculated. Moreover, the authors reported the quantification as Loggenomes/ml, without specifying how this value is obtained starting from the calibration curve that is based on ng of DNA. Please specify your calculations.
-lines 147-149 - here, as before, the nature of the "control" is not clear, please specify.
- lines 176-177 - here, as before, the nature of the "control" is not clear, please specify. Moreover, it should be specified what the authors means with "increase". Did the probiotic growth during the in vitro GI simulation? By quantifying DNA by qPCR the authors cannot determine the viability of the probiotic strain. Its viability across the GI simulation could have been easily quantified by plating on semi-selective media followed by a strain-specific qPCR carried out on the cultivated biomasses. In this context, the sample in Fig 1 that is not significantly different from the control (please define what control is) could represent dead cells that did not undergo lysis.
-Abstract and paragraph 3,2 - The authors wrote in the abstract that "After simulated digestion ... and the observed changes to the metabolite profile were consistent with the those expected from HN019 or the food matrix used". These sentence is too speculative because samples that showed the higher values of Loggenomes/ml of probiotics did not show the highest values of acetic acid or propionic acid, therefore it not clear at all why the "metabolite profile" were consistent with those expected from HN019! From the data it seams that SCFA profile is more related to the food matrix considering that a similar amount of probiotic among all samples (except yogurt). Moreover, line 267-269, the authors themselves underlined that the "intestinal microbiota" is playing a role in SCFA and BCFA amount, and not the probiotic strain.
-discussion section, lines 234-245 - here the authors should specify that the qPCR assay developed cannot distinguish between live and dead cells. Therefore considering that the probiotic definition is based on consumption of live bacteria, this is an important issue to be underlined. As a consequence, the statement that the food matrix had no effect on delivery of HN09 should be implemented with comment of possible effects on cell viability, that authors did not investigate.
-Conclusion - the authors should specify also in this section that they did not measure the survival of the probiotic strain and therefore, that the matrix had no effect of cell delivery but the effect on cell survival should be further investigated. Also, when authors wrote on the "impact on the metabolite profile" by the matrix, the role of the microbiota should be highlighted.
-line 297-300- this sentence is speculative and should be deleted. Several additional factors can affect a clinical trials is probiotics are administered as freeze-dried culture or they are included in food matrixes.
Author Response
The authors would like to thank the esteemed reviewer for their considered comments. We have gone over the criticisms and have the following adjustments or responses:
Comment 1 -Table 1 - The authors should specify to what amount of matrix the CFU refers. Are they CFU/g or total CFU in a defined amount of matrix.
Response 1:
The authors have made an addition to Line 95: ..of matrix
‘, inoculated with probiotic to the total amounts shown in Table 1,’
Comment 2 - lines 107 -110 - Here a definition of "control" is reported. However it not clear which data are obtained from the "control". Is this "control" the same used in Figure 1? Please specify.
Response 2:
The authors have made 2 additions to help clarify what the control value was:
Line 108 has been altered to
‘In addition to the matrices test a control not containing any probiotic was digested in the same manner. All downstream analyses were performed on all test conditions and the control, this was to normalize for donor microbiota variations.’
Figure 1 caption line 183 was altered to:
… and the colon. ‘Control was a colon simulator subunit that had no test substance added to its feed vessel, controlling for donor microbiota variation per simulation’
Comment 3 - lines 139-141 - The authors describe how they obtained the limit of quantification of the qPCR assay developed. Hovewer, the value of this limit is not reported. In addition the authors arbitrary fixed the limit of detection as half of the value of the limit of quantification, whereas it should be calculated. Moreover, the authors reported the quantification as Loggenomes/ml, without specifying how this value is obtained starting from the calibration curve that is based on ng of DNA. Please specify your calculations.
Response 3:
The authors changed Line 140:
‘a theoretical limit of detection (5,37 log genomes/ml) was set at half the limit of quantification, which was the last standard of 10 fg. This limit of detection value was used instead of zero.’
As it pertains to the arbitrary nature of setting the LOD as half the LOQ, this was used as a conservative approximation for other methods of determining an LOD. Which enables statistical analysis of our data set.
Towards the final point in this comments the author have added to Line 142:
‘The unit of Loggenomes/ml was determined from the original qPCR value in genomes/ng of DNA, correlating it through the DNA concentration in ng of DNA/µl and then finally taking into account the dilution factors associated with sample handling post simulation (i.e bead beating and DNA extraction).’
Comment 4 - lines 147-149 - here, as before, the nature of the "control" is not clear, please specify.
Response 4:
The authors feel that by clarifying the definition of control as in comment 2, we also have addressed this comment.
Comment 5 - lines 176-177 - here, as before, the nature of the "control" is not clear, please specify. Moreover, it should be specified what the authors means with "increase". Did the probiotic growth during the in vitro GI simulation? By quantifying DNA by qPCR the authors cannot determine the viability of the probiotic strain. Its viability across the GI simulation could have been easily quantified by plating on semi-selective media followed by a strain-specific qPCR carried out on the cultivated biomasses. In this context, the sample in Fig 1 that is not significantly different from the control (please define what control is) could represent dead cells that did not undergo lysis.
Response 5:
As it regards the issue of defining what the control was throughout the experiments the authors feel they have clarified this in comment 2.
To the point about the use of “increase” was aptly spotted. We meant to say that the qPCR level was statistically “higher” than its relative control, the control further being explained in comment 2. We will change Line 177:
‘All the matrices, except for the dairy yoghurt, were statistically higher than the control (Figure 1).’
This issue of bacterial viability a strong one. The authors in Lines 236-239 make note that the qPCR DNA recovery is only reflects an approximation of intact bacteria. Further, we refer to a pepsin mediated DNA degradation that strengthens our approximation, pepsin being a key component to our upper GI digestion buffers and therefore any exposed DNA coming from dead or lysed bacteria would have likely been degraded. This is not the same as viability and so we continue to use the word approximation.
The reviewer wisely points out that semi-selective media is a powerful microbiological tool and would have been a valuable addition.
Comment 6 - Abstract and paragraph 3,2 - The authors wrote in the abstract that "After simulated digestion ... and the observed changes to the metabolite profile were consistent with the those expected from HN019 or the food matrix used". These sentence is too speculative because samples that showed the higher values of Loggenomes/ml of probiotics did not show the highest values of acetic acid or propionic acid, therefore it not clear at all why the "metabolite profile" were consistent with those expected from HN019! From the data it seams that SCFA profile is more related to the food matrix considering that a similar amount of probiotic among all samples (except yogurt). Moreover, line 267-269, the authors themselves underlined that the "intestinal microbiota" is playing a role in SCFA and BCFA amount, and not the probiotic strain.
Response 6:
The authors agree with the above comment, and have removed HN019 from the sentence so it now reads:
Line 18: …observed changes to the metabolite profile were consistent with those expected from the food matrix used.
Bringing it in alignment with lines 267-269
Comment 7 - discussion section, lines 234-245 - here the authors should specify that the qPCR assay developed cannot distinguish between live and dead cells. Therefore considering that the probiotic definition is based on consumption of live bacteria, this is an important issue to be underlined. As a consequence, the statement that the food matrix had no effect on delivery of HN09 should be implemented with comment of possible effects on cell viability, that authors did not investigate.
Response 7:
The authors have made of modifications to Line 235:
‘…DNA recovery, that we have used as an approximation for transit through the human gastrointestinal tract, we acknowledge that by definition probiotics must be alive and that recovered DNA is not necessarily an indicator of viability. However, ...
Comment 8 - Conclusion - the authors should specify also in this section that they did not measure the survival of the probiotic strain and therefore, that the matrix had no effect of cell delivery but the effect on cell survival should be further investigated. Also, when authors wrote on the "impact on the metabolite profile" by the matrix, the role of the microbiota should be highlighted.
Response 8:
The authors have added a few final sentences as suggested by the reviewer (i.e):
Line297 add ..simulated GI Tract.
‘All together, this work showed the delivery matrix had no appreciable effect on probiotic delivery through simulated human digestion. Though this work focused on probiotic DNA recovery as an approximation for intact bacteria the field of molecular biology has expanded greatly in recent years. Future work could adopt methods involving digital PCR and a viability dye to further establish the link between PCR results and viable bacteria [Kiefer et al 2020].’
Comment 9 - line 297-300- this sentence is speculative and should be deleted. Several additional factors can affect a clinical trials is probiotics are administered as freeze-dried culture or they are included in food matrixes.
Response 9:
The authors agree and the final sentence has been removed and replaced with the sentences suggested by the reviewer in comment 8
Reviewer 2 Report
The manuscript deals with the effect of delivery matrix on Bifidobacterium animalis subsp. lactis HN019 survival through simulated human digestion using modern molecular approaches, comparing common and novel matrices.
After a detailed revision, I noted that the study is well planned, structured, with the results adequately presented and discussed.
Author Response
The authors would like to warmly thank the honorable reviewer for their time and consideration in the peer review process. We were pleased to have submitted a satisfactory manuscript that could hopefully contribute to the field.
Round 2
Reviewer 1 Report
The manuscript was revised accordingly with reviewer's comments. However, some points still need to be addressed.
-lines 108 - 110 - Based on this "control" definition are still not clear the data presented in Figure 1. If the control is a sample "..not containing probiotic", why did the authors quantified the HN019 strain in the control?
Author Response
Thank you for keenly spotting another area that needed improving in our manuscript.
The qPCR assay we used in this study, while not perfect, was quite "strain-specific" as we detailed. However, this is not normally the case. In general we run assays that have background coming from the donor's microbiota and therefore we use the control as a way to set a baseline and the show an increase from baseline for the probiotics or other general microbiota shifts. In essence it functions like a blank that we then compare our test values against.
The lines now read:
"In addition to the matrices a control not containing probiotic was digested in the same manner. All downstream analyses were performed on all test conditions and the control, this was to normalize for donor microbiota variations and any possible background, in particular for any other HN019-like organisms."